# Validation of the Pain and Sensitivity Reactivity Scale in Neurotypical Late Adolescents and Adults

**DOI:** 10.3390/ejihpe15050080

**Published:** 2025-05-13

**Authors:** Agustín Wallace, Lidia Infante-Cañete, Agustín Ernesto Martínez-González, José Antonio Piqueras, Silvia Hidalgo Berutich, Tíscar. Rodríguez-Jiménez, Pedro Andreo-Martínez, Beatriz Moreno-Amador, Alejandro Veas

**Affiliations:** 1Department of Developmental and Educational Psychology, Faculty of Psychology, University of Malaga, 29010 Malaga, Spain; awallace@uma.es (A.W.); lidiainfante@uma.es (L.I.-C.); shidalgo@avanza-online.es (S.H.B.); 2Department of Developmental Psychology and Didactics, University of Alicante, 03690 Alicante, Spain; 3Department of Health Psychology, Miguel Hernández University of Elche, 03202 Elche, Spain; jpiqueras@umh.es (J.A.P.); bmoreno@umh.es (B.M.-A.); 4Area of Personality, Assessment and Psychological Treatments, Department of Psychology and Sociology, Faculty of Social and Human Sciences, University of Zaragoza, 50009 Zaragoza, Spain; trodriguez@unizar.es; 5Department of Agricultural Chemistry, Faculty of Chemistry, University of Murcia, 30720 Murcia, Spain; pam11@um.es; 6Department of Developmental and Educational Psychology, Faculty of Psychology, University of Murcia, 30720 Murcia, Spain; alejandro.veas@um.es

**Keywords:** sensory response, sensitivity reactivity, sensory hyporeactivity, sensory hyperreactivity, sensory over-responsivity, sensory under-reactivity, pain, adolescents

## Abstract

Background: In recent years, there has been an increased interest in studying sensory responses to stimuli in both clinical and non-clinical populations. Sensory reactivity has been linked to restrictive and repetitive behaviors. However, few instruments have been designed to assess the dimensions of sensory hyporeactivity and pain in the general population. Methods: The psychometric properties of the Pain and Sensitivity Reactivity Scale (PSRS) were analyzed in a non-clinical sample of 1122 adolescents and adults (mean age = 22.39, SD = 7.32). Results: The PSRS exhibited excellent psychometric properties, and three first-order factor models were confirmed. The sensory hyperreactivity subscales were highly correlated with the sensory over-responsivity scales, whereas a moderate correlation was found between sensory hyperreactivity measured via the PSRS and OCI-R subscales. Furthermore, sensory hyporeactivity and hyperreactivity appear to be moderately and positively correlated. Differences were observed as a function of gender and age. Conclusions: PSRS may be a reliable measure for analyzing pain and sensory reactivity in neurotypical populations. Future research should include clinical samples and multiple informants.

## 1. Introduction

Accurate understanding and assessments of pain and sensory reactivity are essential to properly address pain management in various populations, including neurotypical adolescents. Late adolescence, a stage marked by significant physical, emotional, and cognitive changes, is a critical period during which pain experiences can have lasting consequences for well-being and quality of life ([21]). In this context, reliable and validated measurement tools to assess pain sensitivity and reactivity in this population are paramount.

Altered sensory responsiveness refers to impairment when modulating outputs to several forms of sensory stimuli, including visual, auditory, tactile, odor, taste, and proprioceptive stimuli. Thus, affected individuals face challenges when regulating and organizing the type and intensity of behavioral responses to sensory inputs to match environmental demands. Sensory responsiveness can be classified into three patterns: sensory hyperreactivity/sensory over-responsivity (SOR), sensory hyporeactivity/sensory under-responsivity (SUR), and sensation seeking ([41]). In addition, altered sensory responsivity is included as a component of restricted and repetitive behaviors (RRBs) pertaining to autism spectrum disorder (ASD), as outlined in the DSM-5-TR ([3]). Several studies have reported a relationship between RRBs and emotional states in individuals with ASD and obsessive–compulsive disorder (OCD) given that RRBs have been linked to anxiety and stress (e.g., [25]; [44]; [51]; [54]; [33], [35]). In addition, SOR-related issues in such individuals are more intense than those experienced by healthy individuals with SOR ([37]). This provides further evidence of the causal link between neurological disorders and SOR while also pointing to its transdiagnostic nature ([27]).

As previously mentioned, sensory reactivity has been extensively studied in clinical populations, such as those with ASD and OCD, where a significant prevalence of SOR symptoms has been demonstrated ([10]; [29]). However, studies on sensory reactivity in neurotypical populations are scarce. Studies in non-clinical populations are usually limited and often focus on comparing sensory thresholds or pain sensitivity between the sexes ([23]; [7]). The lack of research on neurotypical individuals hinders a comprehensive understanding of sensory variability in the general population and highlights the need to develop measurement instruments applicable to this population ([43]). Screening studies in the general population have found a strong association between symptoms of OCD and SOR ([11]; [43]; [58]). Specifically, [16] ([16]) found SOR to be moderately correlated with the checking (0.47), obsessions (0.42), and ordering (0.49) subscales of the obsessive-compulsive inventory—parent version and the OCD subscale of the SCARED-R (0.42). Similarly, another recent study found that the touch factor of SOR was the most strongly correlated factor pertaining to the OCD spectrum (symmetry/ordering = 0.43; contamination/cleaning = 0.41; thoughts/checking = 0.36). In contrast, the taste factor of the SOR was more weakly correlated with the thoughts/checking subscale (0.26) ([43]).

Recent findings indicate that SOR, abdominal pain, and anxiety are interrelated in clinical and non-clinical populations ([18]; [38]; [40], [39]; [61]). In this respect, functional abdominal pain disorders are common in 3–16% of the general population ([61]). In addition, several studies have pointed to a relationship between abdominal pain, selective dietary patterns, emotional instability, and gut dysbiosis in neurodevelopmental disorders such as ASD, all of which seem to be related to the gut–brain microbiota axis ([4], [5], [6]; [30], [31], [32]).

With regard to possible sex differences in sensory reactivity, disparate outcomes have been reported as a function of the sample under study (e.g., ASD, OCD, or neurotypical samples). Various studies in non-clinical populations have reported that males exhibit lower tolerance and sensory thresholds for heat, cold, and pressure than did females ([23]; [49]). A recent study with a sample of 930 typically developing adults found that women reported significantly higher scores on somatosensory and pain subdomains than did men. Nonetheless, no differences were found in visual and auditory sensitivity in this non-clinical population ([7]). However, SOR symptoms have distinct sex-based neural correlates in individuals with ASD. Specifically, in males with ASD, the association between SOR and increased connectivity was strongest in relation to attentional networks and primary sensory networks, whereas in females with ASD, SOR was more strongly related to increased connectivity between attentional networks and the prefrontal cortex. This provides a privileged snapshot of the different symptom patterns exhibited by males and females with ASD ([17]). Furthermore, a higher prevalence of self-reported sensorimotor symptoms has been found in females with level 1 autism than in males ([45]). However, in contrast to ASD, boys and girls with OCD do not seem to exhibit similar reactivity to sensory stimuli ([62]). In consideration of the full body of available evidence, sex differences regarding SOR may be due to the heterogeneity characterizing mental disorders, with OCD, in particular, exhibiting a high degree of heterogeneity ([50]).

Different scales have been used to analyze sensory symptoms in both clinical and non-clinical populations. Self-report measures include, the short sensory profile 2 (SSP-2; [20]), the sensory experiences questionnaire version 3.0 (SEQ-3.0; [8]), the Glasgow sensory questionnaire (GSQ; [53]), the sensory over-responsivity scales (SORS; [22]) and the sensory over-responsivity inventory (SensOR; [56]), amongst others. However, not all measures are available in a self-report format. Instead, many instruments are designed to be completed by primary caregivers as key informants (e.g., SSP-2). Notably, some of these scales have not been previously validated in neurotypical populations following a planned approach to address previously established objectives regarding the psychometric validation of instruments. Thus, new self-report instruments, including those for sensory responses and pain, must be developed for use in clinical and non-clinical populations. Pain (abdominal pain) is an independent variable related to SOR and anxiety, meaning that it may share some common underlying mechanisms ([40], [39]). In this sense, screening for these factors (pain and reactivity response) may help with the development of an integrative treatment approach. Therefore, the development of these new instruments must incorporate pain as a determining factor ([46]). The development of self-report instruments to measure sensory reactivity and pain is necessary to analyze sensory profiles in the general population and to understand the way in which their symptoms differ from those found in clinical populations.

Thus, the present study aims to examine the psychometric properties of a new instrument, the Pain and Sensitivity Reactivity Scale (PSRS), administered to a community-based sample of Spanish adolescents and adults. To this end, the following specific objectives were established: (i) conduct an exploratory factor analysis; (ii) examine the factor structure of the tool; (iii) assess its internal consistency and test–retest reliability; (iv) examine its convergent (sensory over-responsivity scales) and discriminant validity (obsessive-compulsive inventory—revised); and (v) explore the outcomes, both overall and for individual subscales as a function of gender and age.

## 2. Materials and Methods

An instrumental study was conducted in Spain to develop and examine the psychometric properties of the PSRS in a sample of neurotypical late adolescents and adults. The COSMIN methodology for the development of the PSRS has been used ([42]; [60]).

### 2.1. Measurement

#### 2.1.1. Sensory Over-Responsivity Scales (SOR-Scales; [22])

The SORS assesses sensory hyperreactivity to auditory, tactile, visual, olfactory, and taste stimuli. The present study used a version adapted from a version administered to a general community sample in a survey study ([58]). It consists of rating scales that examine distress and impairment of both auditory and tactile over-reactivity ([22]). Each SORS subscale contains four items that are rated on a five-point scale ranging from 0 to 4, with overall scores ranging from 0 to 80. The total scores for each subscale are calculated individually and range from 0 to 16, with higher scores indicating greater severity. The internal consistency of the SORS and its subscales, examined according to Cronbach’s alpha, has been found to be strong in a sample from the United States (SOR-overall = 0.93; SOR-hearing = 0.89; SOR-touch = 0.88; SOR-smell = 0.90; SOR-sight = 0.94; SOR-taste = 0.88) and a sample from Spain (hearing = 0.89; touch = 0.86; smell = 0.91; sight = 0.90; taste = 0.86) ([43]).

#### 2.1.2. Obsessive-Compulsive Inventory—Revised (OCI-R; [24])

The OCI-R is an 18 item self-report questionnaire that assesses obsessive–compulsive symptom severity using a five-point Likert scale ranging from 0 (not at all) to 4 (very much). The OCI-R comprises six factors that represent the following symptom domains: checking, ordering, neutralizing, washing, obsessing, and hoarding. Each factor is composed of three items, with overall scores ranging from 0 to 12. Overall, the measure has exhibited good internal consistency in samples from a number of different countries, with Cronbach’s α outcomes ranging from 0.81 to 0.95 ([26]; [36]; [48]).

#### 2.1.3. Pain and Sensitivity Reactivity Scale (PSRS)

The PSRS is a tool that evaluates reactivity to pain and sensory reactivity according to 50 items. It is composed of three dimensions, namely, pain, sensory hyporeactivity, and sensory hyperreactivity. The items are rated on a four-point Likert scale ranging from 0 (behavior does not occur) to 3 (behavior occurs and is a severe problem). Both the hyposensitivity and hypersensitivity dimensions comprise tactile, olfactory, visual, gustatory, and auditory items. In addition, the PSRS includes a pain reactivity domain which comprises seven items. The PSRS was elaborated on based on theoretical requisites conceived by [41] ([41]), which characterizes sensory modulation disorders according to three patterns (hyper-response, hypo-response, and sensory seeking) as proposed nosology for diagnosis. Two version of the PSRS are available. The first is a version administered for completion by caregivers and professionals, whilst the second is a self-report version. The caregiver version of the PSRS has shown excellent internal consistency in samples with ASD (pain α = 0.83; broad sensory hyporeactivity α = 0.90; broad sensory hyperreactivity α = 0.93) ([34]). The self-report version was used in the present study.

### 2.2. Development and Content Validity of the New Instrument

The purpose of the PSRS is to measure sensory reactivity and pain in individuals with typical development and individuals with ASD. The first step in the present study was to validate the PSRS in a non-clinical population.

Elaboration of the PSRS was based on the theoretical model conceived by [41] ([41]) with regard to SOR and SUR. Similarly, the PSRS is grounded in existing evidence pertaining to pain from the field of neuroscience ([28]), together with studies arguing that pain is an important variable for explaining sensory reactivity ([40], [39]). The structure of the PSRS adheres to the same parameters as other validated scales that measure symptom severity (e.g., RBS-R).

The PSRS was developed by a multidisciplinary team. Finally, 50 real situations involving SOR and SUR that were reported by families were selected. With regard to the pain factor, initially, different origins of physical pain were recorded according to records obtained from pediatric and psychiatric services. This produced a total of 32 items related to different physical and medical situations underlying pain. Three doctors (two pediatricians and one psychiatrist) screened the gathered situations, producing a final set of seven items. A large number of items were found to be redundant with regard to at least one of the reported pain-related situations. Consequently, all items were grouped into a single factor, denominated as the pain factor. Subsequently, examples comprised by each item were reviewed. This process was carried out by a pediatrician, a neuro-psychologist, and a psychologist. The three experts were all PhDs. Two of them were university professors in the Faculty of Psychology and were also experts in research methods. The other expert has more than twenty years of experience in medical assessments. There were no changes in the items between the caregiver version of the PSRS scale and the self-report scale. The grammar was adapted to the first-person singular for each of the items in the self-reported version of the PSRS. Furthermore, the protocol of [19] ([19]) was applied to all the experts so that they could evaluate the items. The judges rated the relevance of each item on a 5-point scale (1: low degree of item clarity; 5: high degree of item clarity) and found a consensus. The means for item clarity were calculated, considering items with a score of 3 or more, out of 5 ([2]), as adequate. Items were considered to have an adequate degree of relevance if the V-index was above this cut-off point and the 95% confidence interval did not include the value 0.70 ([1]). The mean item clarity scores were above 3.40 in all cases, indicating that the experts considered the items to be clearly worded. Likewise, the clinical opinion of the experts was taken into consideration to indicate the most frequent items in consultation. Subsequently, the self-report version of the instrument was administered to ten young adults to measure their level of understanding in relation to the items. Finally, minor adjustments were made to ensure comprehension and clarity of items.

### 2.3. Procedure

The participants completed all study procedures during regular timetabled lessons. Young adults from different universities across Spain participated in the study (Alicante, Elche, Murcia, Malaga, and Zaragoza). Late adolescents who were not yet in university (16 years old) were selected from two secondary education centers in Alicante and Murcia. All participants were residents of Spain. The survey was filled out online. The reporting assessment protocol was individually applied using the online survey tool LimeSurvey (LimeSurvey GmbH Hamburg, Germany). The assessment protocol was only written in Spanish. Appropriate instructions were provided for the completion of each inventory. A researcher remained in the classroom throughout the administration to assist students who experienced difficulties. The tests were administered by experienced psychologists who provided instructions and individual assistance to all students who needed it. Approximately 20 min was required for completion of all scales. One month following the initial administration (T0), the data collection procedures were repeated with a random sample of 124 college students who had participated in the first round of data collection (T1). The participants did not receive any financial compensation for their participation in the present study. The survey was conducted between November 2020 and December 2021, with a time-lapse of around one month between T0 and T1.

### 2.4. Data Analysis

First, the structural validity of the PSRS was evaluated. In short, the PSRS comprises a pain scale made up of seven indicators/items, a broad sensory hyporeactivity scale with five subscales (4–6 items each) and a broad sensory hyperreactivity scale, also, with five subscales (4–5 items each). With regard to factorial structure, the five first-order sensory hypo- and hyperreactivity factors were defined as indicators of the higher, second-order hypo- and hypersensitivity scales, which were then modeled as indicators of a broad third-order sensory reactivity scale in accordance with the proposed theoretical model

An exploratory factor analysis was performed to examine the factor structure of the scale using SPSS 24.

A confirmatory factor analysis was conducted using EQS 6.2 software (Multivariate Software, Inc., Temple City, CA, USA). Current guidelines for good model fit suggest that comparative fit (CFI) and Tucker–Lewis (TLI) indices greater than 0.90 indicate good model fit, alongside root mean square error of approximation (RMSEA) values lower than 0.05. In addition, chi-square change (Δχ2) values were used, alongside CFI, to compare the goodness of fit between models whose values were greater than 0.90 ([12]). Due the ordinal nature of the item, we used Robust Weighted Least Square (WLSMVS) as the estimation method. To ensure the stability of the results, the total sample was divided into two samples through random sampling.

Temporal stability and correlations between the PSRS subscales were examined using IBM SPSS statistics version 24 (IBM Corp., 2016, Armonk, NY, USA). For temporal stability, correlations exceeding 0.70 have been suggested as acceptable for group comparisons ([47]), whereas Cohen’s criteria ([15]) were used to assess the magnitude of the relationships exhibited between different variables. In this sense, correlations above 0.50 were considered high, whilst correlations between 0.30 and 0.49 were considered moderate, and correlations between 0.10 and 0.29 were considered low.

Reliability was assessed in line with multiple indicators. Given the multidimensional nature of the instrument, Ordinal Cronbach’s alpha and Ordinal McDonald’s omega values were produced ([52]). Values between 0.80 and 0.90 are considered acceptable for both of these indices ([55]). In order to examine convergent validity, a Pearson correlation analysis was conducted to reveal associations between the overall and individual subscale PSRS scores and the overall and individual subscale SORS scores. Finally, discriminant validity was examined according to Pearson correlations between the overall and individual subscale PSRS scores and OCI-R subscale scores.

### 2.5. Ethical Considerations

The present study was approved by the ethics committee (reference number: UA-2019-10-04, approval date: 27 March 2020). The adult participants provided written informed consent, whilst consent for the participation of minors was provided by their parents or legal guardians in accordance with the Declaration of Helsinki.

## 3. Results

### 3.1. Characteristics of the Sample

An incidental sample of 1122 Spanish adolescents and adults was recruited to participate in the present study, with 294 being male and 818 being female. An overall response rate of 99.1% was achieved among the participants. Ten adolescents did not want to participate and were excluded from the study (three from Alicante, four from Andalusia, and three from Murcia). The mean age was 22.39 (SD = 7.32), with 48% of the sample being under 19 years old. The sample consisted of 1122 (T0) and 124 (T1) Spanish adolescents and adults aged 16–68 years who completed all survey measures. Table 1 presents the participant’s sociodemographic characteristics.

### 3.2. Inter-Item Correlations

Outcomes pertaining to the correlation matrix revealed that no items produced correlations above 0.85, with the greatest correlation between two items (48 and 49) being 0.65.

### 3.3. Exploratory Factor Analysis

An exploratory factor analysis was performed employing principal axis factoring and oblimin rotation, extracting factors with eigenvalues greater than one. Scale dimensionality was examined, firstly, in accordance with a three-factor model and, secondly, in accordance with a model composed of 11 factors. The latter model produced better outcomes results, with Bartlett sphericity outcomes of χ^2^ (1225) = 17,263.90 (*p* < 0.001), KMO of 0.925, and less than 1% redundant residue.

### 3.4. Confirmatory Factor Analysis

Goodness of fit indices indicated that three different correlated models comprising (i) three first-order factors, (ii) three factors, and (iii) eleven first-order factors grouped into two second-order correlated factors modeled as indicators of a broad third-order sensory reactivity scale fitted the data acceptably. The CFI, TLI, and GFI values were all equal to or greater than 0.90, whilst the RMSEA values were less than 0.05 (see Table 2).

Statistically significant differences emerged between the model comprising three correlated factors and the model comprising eleven first-order factors grouped into two second-order correlated factors (Δχ2 = 1294.11; Δdf = 3; *p* =< 0.001), with the second model exhibiting higher CFI values. The results obtained using the split samples show similar results, which indicates stability of the fitted model (see Table 3).

All items fit well with each factor, with the exception of item 17 pertaining to olfactory sensory hyporeactivity (factor 3). In addition, factorial invariance as a function of gender and age was examined (item 17 removed), with outcomes revealing that all examined models equally fit the gathered data (see Table 4).

### 3.5. Inter-Scale Correlations

As shown in Table 5, all PSRS subscale scores were moderately to highly correlated with overall PSRS scores. Correlations between the different examined scales were of low to moderate strength.

### 3.6. Reliability Measures

The reliability (Ordinal Cronbach’s alpha and Ordinal McDonald’s omega) outcomes pertaining to the PSRS are presented in Table 6. As can be observed, the majority of the reliability indices corresponding to the overall PSRS and its individual subscales indicate high reliability. The weakest outcomes were produced in relation to the visual hyperreactivity scale compared with those for the other subscales.

With regard to test–retest reliability, correlations for all scales were statistically significant (*p* < 0.01). When examining the overall PSRS, a strong four-week test–retest correlation (*r* = 0.86) was produced. Furthermore, high strength coefficients were produced for all PSRS subscales, with the only exception being the visual hyperreactivity scale (*r* = 0.52), which produced a weaker correlation than all the other subscales.

### 3.7. Convergent and Discriminant Validity

As presented in Table 7, correlations between the PSRS hyperreactivity scale and measures selected to examine its convergent validity (e.g., SOR-touch, SOR-smell, SOR-sight, SOR-taste, and SOR-hoarding) were of greater strength than those produced with measures selected to examine its discriminant validity (e.g., OCI-R), with correlations pertaining to the former ranging from 0.60 to 0.59, compared with 0.28 (PSRS-hyper-taste and OCI-R-checking) to 0.48 (PSRS-hyper-tactile, OCI-R-ordering, and OCI-R-obsessing) for the latter. In contrast, correlations between the sensory hyporeactivity subscale and all OCI-R subscales were between 0.22 (OCI-R-hypo-taste and OCI-R-washing) and 0.38 (OCI-R-hypo-auditory and OCI-R-hoarding). The PSRS pain subscale examined was only weakly correlated with the SORS and OCI-R, with the weakest correlation being produced with regard to the OCI-R. In most cases, correlations indicating convergent validity of the overall PSRS were statistically stronger than those indicating its discriminant validity, though evidence generally supports the convergent and discriminant validity of the PSRS overall. Furthermore, the correlations produced between the PSRS subscales and scales selected to examine convergent validity (e.g., SOR-touch, SOR-smell, SOR-sight, SOR-taste, and SOR-hearing) were higher with regard to the sensory hyperreactivity items, ranging from 0.50 to 0.66, than with regard to the sensory hyporeactivity items, ranging from 0.36 to 0.40.

### 3.8. Invariance as a Function of Gender

In order to determine factorial invariance as a function of gender, the configural (M1), metric (M2), strong (M3), and strict (M4) invariances were estimated for the 11-factor model ([13]). Firstly, the examination of the same PSRS structure applied to both genres (M1) produced good outcomes (RMSEA = 0.037 [90% CI = 0.035–0.039]; CFI = 0.980). The outcomes reveal that the model adequately fit the data in both groups. Secondly, taking M1 as a reference, M2 was tested employing a model in which the factor loadings were fixed within both groups. The outcomes demonstrated that M2 also provides a good fit to the obtained data (RMSEA = 0.044 [90% CI = 0.042–0.046]; CFI = 0.973). When comparing the model fit between M2 and M1, no significant differences were observed (ΔCFI = 0.007; ΔRMSEA = 0.007). These findings suggest that the factor loadings are invariant between groups defined according to gender. Thirdly, M3 was examined, in which the factor loadings and intercepts were defined to be equal in both groups. The outcomes indicated that M3 was also a good fit to the obtained data (RMSEA = 0.041 [90% CI = 0.039–0.043]; CFI = 0.975). When comparing the model fit of M2 and M3, no significant differences were evident (ΔCFI = 0.002; ΔRMSEA = 0.003). Thus, the hypothesis that model intercepts are invariant to gender is accepted. Finally, M4 was analyzed, in which the factor loadings, intercepts, and residuals were fixed and held the same in both groups. Again, adequate outcomes were produced (RMSEA = 0.041 [90% CI = 0.039–0.043]; CFI = 0.975). When compared with M3, the extent to which the fit indices changed between models (ΔCFI = 0.000; ΔRMSEA = 0.000) provides further empirical support of strict invariance. Taken together, these outcomes indicate measurement invariance of the PSRS as a function of gender with its factor structure being maintained in both males and females.

### 3.9. Invariance as a Function of Age

In order to determine factorial invariance as a function of age (adolescents versus adults), the configural (M1), metric (M2), strong (M3) and strict (M4) invariances were estimated for the 11-factor model ([13]). Firstly, the factor structure of the PSRS was examined between both age groups (M1), producing a good outcome (RMSEA = 0.042 [90% CI = 0.040–0.044]; CFI = 0.975). The outcomes demonstrate that the model presented an adequate fit to the data in both groups. Secondly, taking M1 as a reference, M2 was examined, in which the factor loadings were fixed and held constant in both groups. The outcomes revealed that M2 was a good fit to the data (RMSEA = 0.047 [90% CI = 0.046–0.049]; CFI = 0.970). When comparing M2 with M1, no significant changes in model fit were observed (ΔCFI = 0.005; ΔRMSEA = 0.005). These findings suggest that the factor loadings are invariant between groups. Thirdly, M3 was evaluated, in which the factor loadings and intercepts were held the same in both groups. The outcomes indicated that M3 showed adequate fit indices (RMSEA = 0.044 [90% CI = 0.043–0.046]; CFI = 0.971). When comparing M2 with M3, no significant changes were evident (ΔCFI = 0.001; ΔRMSEA = 0.003), and so, the hypothesis that model intercepts are invariant between age groups is accepted. Finally, M4 was analyzed with the factor loadings, intercepts, and residuals being held constant in both groups. Again, adequate fit indices were produced (RMSEA = 0.044 [90% CI = 0.043–0.046]; CFI = 0.971). When comparing the fit indices for M3 and M4, the extent of change (ΔCFI = 0.000; ΔRMSEA = 0.000) provides empirical support of strict invariance. Taken together, all outcomes indicate factorial invariance of the PSRS, with its factor structure being maintained in the two different examined age groups.

### 3.10. Descriptive Statistics of the PSRS: Latent Mean Differences

In order to estimate differences between the groups as a function of both gender and age, males and adolescents, respectively, reference groups were established (means fixed at zero). Critical ratios (CRs) were calculated to estimate whether differences as a function of gender and age were statistically different from zero. Critical ratios larger than 1.96 indicate that the mean value recorded for the comparison group is greater than that in the reference group.

Appendix A presents the differences in PSRS scores according to gender. The outcomes reveal higher levels of visual (CR = −4.18), taste (CR = −4.05), auditory (CR = −5.55), and overall sensory hyporeactivity (CR = −4.10) in men than in women, with the magnitude of differences being small (*d* between 0.19 and 0.34). However, women reported higher values of tactile (CR = 2.70), olfactory (CR = 2.35), and overall sensory hyperreactivity (CR = 2.03) and pain (CR = 6.50), with the magnitude of differences being small (*d* between 0.11 and 0.17).

With regard to the differences in PSRS scores as a function of age, the outcomes reveal statistically significant differences in sensory hyporeactivity and hyperreactivity, with both being greater in late adolescents than in adults (see Appendix A). Specifically, adolescents reported higher scores of overall hyperreactivity (CR = −2.16) and tactile (CR = −2.83), olfactory (CR = −2.19) and taste (CR = −2.55) hyperreactivity. Furthermore, adolescents reported higher overall hyporeactivity scores (CR = −4.10), alongside higher tactile (CR = −10.48) and auditory (CR = −3.89) hyporeactivity. The magnitude of differences was small for the majority of sensory dimensions (*d* between 0.11 and 0.26). The analysis failed to identify differences between adolescents and adults with regard to pain (CR = −0.33).

## 4. Discussion

The present study analyzed the psychometric properties of the PSRS in a non-clinical sample of Spanish adolescents and adults. It confirmed that the PSRS is made up of three robust and independent factors. All items presented adequate fit to their respective factors, with the exception of item 17, “I have a hard time perceiving unpleasant odors or bad smells”, which exhibited poor fit. In this sense, the poor fit of this item may be due to the way in which the item was written. This being said, similar items were also written with the same grammatical structure and presented good fit (e.g., item 29). Excellent internal consistency of the PSRS was also found (ω = 0.94), which is in line with that reported in a previous study ([34]). In addition, the test–retest reliability was good (0.86), similar to that reported in a previously conducted study with an ASD population ([34]). In terms of measurement invariance, the factor structure of the PSRS was robust to variations in age and gender.

The initially proposed hypothesis is confirmed in relation to convergent and discriminant validity. Significant correlations, ranging from moderate to strong, were observed between the sensory hyperreactivity subscale of the PSRS and the SORS (r = 0.60 and 0.51). Additionally, similar positive correlations were found between overall hyperreactivity subscales PSRS scores and all individual subscales of the OCI-R. In this sense, these findings coincide with those reported in previous studies which have reported the existence of a relationship between sensory hyperreactivity and OCD symptoms (e.g., [11]; [43]). On the other hand, weaker correlations are observed between the overall sensory hyporeactivity PSRS subscale and the SORS. Furthermore, even weaker correlations were found between the PSRS sensory hyporeactivity subscale and all individual OCI-R subscales. These results indicate that sensory hyporeactivity and repetitive behavior have a smaller association. With regard to pain, the present study indicates a weak relationship between pain and the sensory hyperreactivity and hyporeactivity subscales of the PSRS. This finding suggests that pain is independent of SOR and SUR. Few studies have been conducted to examine these aforementioned relationships in non-clinical populations. However, associations were found between anxiety, SOR, and chronic abdominal pain in relation to ASD, with SOR being a significant predictor of pain onset ([40], [39]). In relation to differences in pain scores according to gender, the findings indicate that females are more sensitive to pain than males. This finding does not coincide with the outcomes reported in previous studies ([23]; [49]) and may be due to differences in the sample size of males and females. In the same way, it must be considered that there may be a bias of the male informant. Due to social desirability, men may not report pain tolerance ([59]). In contrast, no differences in pain were observed according to age. Thus, no clear developmental pattern for pain is evidenced. These findings are consistent with those reported in a recent study examining the relationship between measures used to assess SOR and OCD symptoms in a non-clinical population ([43]). The present findings, therefore, are consistent with the transdiagnostic impact of SOR ([27]). Additionally, the validation analysis conducted in the present study included sensory hyporesponsiveness and pain variables. Thus, the PSRS responds to calls from the research community to include pain and hyposensitivity variables in analyses of sensory reactivity in clinical and non-clinical populations ([7]; [43]).

In relation to gender differences in sensory reactivity, the present findings reveal significantly higher pain and sensory hyperreactivity scores in females than in males. These findings support those reported in another study conducted with a non-clinical population ([7]). In a study conducted by Aykan, pain was considered to be a subdomain of the somatosensorial subscale as opposed to be considering a factor in itself. Similarly, the present findings suggest that no gender differences exist regarding auditory sensory hyperreactivity. This is consistent with findings reported by [7] ([7]). In the same way, a novel contribution of the present work is the emergence of significantly higher levels of sensory hyporeactivity in males than in females. This suggests that a biological sensory pattern or cultural factor may be at play that demands further examination. However, the gender differences identified in the present study may be explained by sample size, with substantially more females being recruited than males.

Sensory reactivity has not been examined in relation to different life stages. Some studies have found that between 5 and 21% of school-age children may have clinical sensory hyperreactivity ([9]; [14]). To the best of our knowledge, the present study is the first to compare sensory reactivity between adolescence and adulthood, revealing that adolescents have higher levels of sensory hyperreactivity (tactile, olfactory, and gustatory) and sensory hyporeactivity (tactile and auditory) than adults. These findings may indicate that developmental patterns exist that underly sensory reactivity in humans, with development rapidly and clearly advancing over the course of the life cycle before leveling off close to adulthood.

Ultimately, the present findings demonstrate that the PSRS has three well-defined first-order factors, with this model showing acceptable fit to the gathered data. Thus, a novel aspect of this new instrument is that the PSRS considers pain as an independent factor, alongside two factors describing sensory reactivity (sensory hyporeactivity and sensory hyperreactivity), which are closely related to restricted behavior, and other variables related to the gut–brain axis ([4], [5], [6]; [30], [31], [32]). However, the present study is not exempt from certain limitations. The first concerns the reliability of data gathered using self-report measures, which is hampered by social desirability. However, self-report measures may better capture sensory reactivity ([57]). Secondly, the cross-sectional nature of this study makes it impossible to make causal inferences. The third limitation is the unequal gender distribution of the sample. Finally, the fourth limitation is that there is not a large sample of participants for the evolutionary periods of late adolescence and early adulthood.

## 5. Conclusions

The validation of the Pain and Sensitivity Reactivity Scale (PSRS) in neurotypical late adolescents and adults could significantly transform practices in health, education, and psychological assessments. In the health sector, this would allow professionals to personalize treatments based on each individual’s pain sensitivity, thereby improving the effectiveness of interventions for conditions such as chronic pain, anxiety, or depression. It would also facilitate earlier detection of mental health issues linked to pain sensitivity, promote preventive interventions, and enhance overall well-being. In the educational context, the PSRS could be used by school counselors to identify students with high pain sensitivity, enabling the implementation of specific strategies, such as stress management programs, contributing to a more inclusive learning environment that addresses individual needs. In psychological assessment, the integration of the PSRS would provide an additional tool for understanding the impact of pain on adolescents’ emotional and behavioral lives, improving diagnostic accuracy and the planning of therapeutic interventions. Overall, validation of the PSRS would enrich practices in these fields, fostering a more holistic and personalized approach to the care and development of adolescents.

## Figures and Tables

**Table 1 ejihpe-15-00080-t001:** Sociodemographic characteristics of the total sample.

Variables	*n* (%)
16–19 years old	539 (48%)
20–68 years old	583 (52%)
Total	1122 (100%)
Gender	
Female	818 (72.9%)
Male	294 (26.2%)
Other	10 (0.9%)
Country/region of birth	
Spain	1079 (96.2%)
Rest of Europe	17 (1.5%)
America	20 (1.8%)
Africa	5 (0.4%)
Asia	1 (0.1%)

**Table 2 ejihpe-15-00080-t002:** Results of the confirmatory factor analysis of the PSRS of the total sample.

Models	χ2	df	RMSEA (CI 95%)	CFI	TLI	GFI
3 correlated factors	4507.28	1172	0.05 (0.05–0.05)	0.96	0.96	0.97
11 factors	2786.15	1120	0.04 (0.04–0.04)	0.98	0.98	0.98
11 + 2 factors *	3074.37	1162	0.04 (0.04–0.04)	0.98	0.98	0.98
Men–Women 11 + 2 factors *	3803.56	2134	0.04 (0.04–0.04)	0.98	0.98	0.97
Adolescents–Adults 11 + 2 factors *	4064.73	2228	0.04 (0.04–0.04)	0.98	0.98	0.97

Notes. * Eleven first-order factors grouped under two second-order factors.

**Table 3 ejihpe-15-00080-t003:** Results of the confirmatory factor analysis of the PSRS with split samples.

Models	χ2	df	RMSEA (CI 95%)	CFI	TLI	GFI
3 correlated factors (s1)	3555.00	1124	0.05 (0.05–0.05)	0.96	0.96	0.96
11 factors (s1)	2549.67	1072	0.04 (0.04–0.04)	0.98	0.98	0.98
11 + 2 factors (s1)	2620.08	1114	0.04 (0.04–0.04)	0.98	0.98	0.98
3 correlated factors (s2)	2339.73	1124	0.05 (0.05–0.06)	0.96	0.95	0.96
11 factors (s2)	1847.48	1072	0.04 (0.04–0.04)	0.98	0.98	0.98
11 + 2 factors (s2)	1943.31	1114	0.04 (0.04–0.04)	0.98	0.98	0.98

Notes. s1 = split sample 1; s2 = split sample 2.

**Table 4 ejihpe-15-00080-t004:** Loadings of the confirmatory factor analysis for the eleven first-order factor model grouped in two second-order factors of the PSRS.

Item PSRS	Factor Loading
Factor 1: Pain	0.55
1. I feel pain or discomfort when I have stomach problems (e.g., constipation, diarrhea, etc.). [Me duele o siento molestias cuando tengo problemas estomacales (p.ej.: estreñimiento, diarrea, etc.)]	0.55
2. It hurts or I feel discomfort when I have inflammation problems (e.g., in the ear, throat, teeth or mouth, etc.). [Me duele o siento molestias cuando tengo problemas de inflamación (p.ej.: en oído, garganta, dental o bucal, etc.)]	0.74
3. It hurts or I feel discomfort when I have eye irritation, conjunctivitis, etc. [Me duele o siento molestias cuando tengo irritación de los ojos, conjuntivitis, etc.]	0.67
4. I feel pain or discomfort when I have a fever. [Me duele o siento molestias cuando tengo fiebre.]	0.82
5. It hurts or I feel discomfort when I have had a fracture or have gone to rehab. [Me duele o siento molestias cuando he tenido una fractura o he ido a rehabilitación.]	0.59
6. It hurts or I feel discomfort when I am punctured for an analysis. [Me duele o siento molestias cuando me pinchan para una analítica.]	0.48
7. It hurts or I feel discomfort when I have fallen or been hit. [Me duele o siento molestias cuando me he caído o dado un golpe.]	0.69
Factor 2: Hypo-Tactile	
8. I prefer very hot or cold water (e.g., taking cold showers or drinking hot tap water, hot soup, etc.). [Prefiero el agua muy caliente o fría (p.ej.: ducharse con agua fría; beber agua caliente del grifo, sopa caliente, etc.).]	0.49
9. I scratch my wounds until they bleed again. [ Me rasco las heridas hasta que vuelven a sangrar.]	0.50
10. I like to dress in tight clothes, socks, and shoes. [Me gusta vestirse con ropa, calcetines y zapatos apretados.]	0.54
11. I squeeze the pen or pencil a lot when writing. [Aprieto mucho el boli o el lápiz al escribir.]	0.53
12. I like to touch things and people. [Me gusta tocar las cosas y las personas.]	0.71
13. I hug people tightly. [Abrazo con fuerza a las personas.]	0.65
Factor 3: Hypo-Olfactory	
14. Certain smells fascinate me. [Me fascinan determinados olores.]	0.85
15. I smell myself, people, and objects. [Me huelo a mí mismo, a las personas y a los objetos.]	0.81
16. I prefer or like intense or strong scents. [Prefiero o me gustan los olores intensos o fuertes.]	0.67
17. I have a hard time perceiving unpleasant odors or bad smells. [Me cuesta trabajo percibir los olores desagradables o malos olores.]	0.12
Factor 4: Hypo-Visual	
18. I am fascinated by moving or rotating objects. [Me fascinan los objetos en movimiento o que giran.]	0.75
19. I prefer or like intense or bright colors. [Prefiero o me gustan los colores intensos o brillantes.]	0.76
20. I am attracted to light and reflections. [Me atrae la luz y los reflejos.]	0.75
21. I have trouble perceiving strong light in front of my eyes (e.g., flashlights, car lights, etc.). [Me cuesta trabajo percibir la luz fuerte ante sus ojos (p.ej.: la luz de una linterna, luces de coches, etc.).]	0.45
Factor 5: Hypo-Taste	
22. I am fascinated by or really like the taste of certain objects or body parts (e.g., sucking on my fingers, rubber objects, etc.). [Me fascina o me gusta mucho el sabor de ciertos objetos o partes del cuerpo (p.ej.: chupar mis dedos, objetos de goma, etc.).]	0.84
23. I like food with strong flavors. [Me gusta la comida con sabores fuertes.]	0.57
24. I like to suck or lick objects, food, etc. [Me gusta chupar o lamer objetos, comida, etc.]	0.85
25. I don’t feel full/satiated after eating a lot. [ No me siento lleno/saciado después de comer mucho.]	0.41
Factor 6: Hypo-Auditory	
26. I am attracted to certain sounds. [Me atraen ciertos sonidos.]	0.74
27. I listen to television or music at a very high volume. [Escucho la televisión o la música a volumen muy alto.]	0.59
28. I like to cause loud noises or sounds. [Me gusta provocar ruidos o sonidos fuertes.]	0.66
29. I find it hard to hear what others say, etc. [Me cuesta trabajo escuchar lo que dicen los demás, etc.]	0.51
Factor 7: Hyper-Tactile	
30. I feel discomfort when touched. [Siento molestia o incomodidad cuando me tocan.]	0.64
31. I feel annoyance or discomfort when I notice skin imperfections (e.g., due to a skin wound, scabs or pimples, etc.). [Siento molestia o incomodidad cuando noto imperfecciones en la piel (p.ej.: por una herida en la piel, costras o granos, etc.).]	0.65
32. I feel discomfort when I touch certain textures (e.g., certain clothing or food). [Siento molestia o incomodidad cuando toco ciertas texturas (p.ej.: ciertas prendas de vestir o alimentos).]	0.67
33. I feel annoyance or discomfort when certain elements or objects that can touch my head or nails come into contact (e.g., showering, getting water on your hair, or having your hair or nails cut). [Siento molestia o incomodidad cuando entran en contacto ciertos elementos u objetos que pueden tocar mi cabeza o uñas (p.ej.: ducharse, que le caiga agua en el pelo, que le corten el pelo o las uñas).]	0.64
34. I feel annoyed or uncomfortable when my favorite or usual clothes are not ready (e.g., fussy about clothes, etc.). [Siento molestia o incomodidad cuando no está preparada mi ropa favorita o habitual (p.ej.: maniático con la ropa, etc.).]	0.63
Factor 8: Hyper-Olfactory	
35. I feel annoyed or uncomfortable when I smell certain odors that other people don’t mind (e.g., colognes, gels, etc.). [Siento molestia o incomodidad cuando huelo ciertos olores que a otras personas no les molestan (p.ej.: colonias, geles, etc.).]	0.71
36. I feel discomfort when I smell certain places (e.g., public toilets, etc.). [Siento molestias o incomodidad cuando huelo ciertos lugares (p.ej.: aseos públicos, etc.).]	0.82
37. I feel discomfort when I smell certain foods. [Siento molestias o incomodidad cuando huelo ciertas comidas.]	0.77
38. I feel discomfort when I smell certain people. [Siento molestias o incomodidad cuando huelo a ciertas personas.]	0.81
Factor 9: Hyper-Visual	
39. I feel discomfort when I see certain colors of food on a plate (e.g., different colors of food, shapes of food, etc.). [Siento molestias o incomodidad cuando veo ciertos colores de alimentos dentro de un plato (p.ej.: diferentes colores de la comida, forma de la comida, etc.).]	0.68
40. I feel annoyed or uncomfortable when I see the physical appearance of some people (e.g., baldness, beard, hair color, etc.). [Siento molestia o incomodidad cuando veo el aspecto físico de algunas personas (p.ej.: calva, barba, color de pelo, etc.).]	0.75
41. I feel annoyed or uncomfortable when I see a change in something or someone (e.g., a certain container of food is not there, etc.). [Siento molestia o incomodidad cuando veo un cambio en algo o alguien (p.ej.: no está un determinado recipiente de comida, etc.).]	0.74
42. I feel annoyed or uncomfortable when I see high-intensity light stimuli or bright light (e.g., car lights, Christmas lights, toy lights, etc.). [Siento molestia o incomodidad cuando veo estímulos luminosos de alta intensidad o la luz brillante (p.ej.: luces de coches, luces de navidad, luces de juguetes, etc.).]	0.53
Factor 10: Hyper-Taste	
43. I feel discomfort from the taste of certain foods, so I only accept some flavors (for example: you prefer very sweet, very salty flavored foods, etc.). [Siento molestias o incomodidad por el sabor de ciertos alimentos, así que solo acepto algunos sabores (por ejemplo: prefiere alimentos con sabor muy dulces, muy salados, etc.). ]	0.80
44. I feel annoyed or uncomfortable with foods with specific textures (e.g., refuses solid foods). [Siento molestia o incomodidad por alimentos con texturas específicas (p.ej., rechaza alimentos sólidos).]	0.75
45. I feel discomfort from foods that are new to me. [Siento molestias o incomodidad por alimentos que son nuevos para mí.]	0.75
46. I feel discomfort when there are changes, however small or subtle, in my favorite foods (e.g., taking another brand of yogurt, etc.). [Siento molestias o incomodidad cuando se producen cambios, aunque sean pequeños o sutiles, en mis alimentos preferidos (p.ej.: tomar otra marca de yogurt, etc.).]	0.70
Factor 11: Hyper-Auditory	
47. I feel annoyed or uncomfortable when I hear certain continuous noises (for example: noise from busy/crowded places, excessive noise in the classroom, etc.). [Siento molestia o incomodidad cuando escucho ciertos ruidos continuos (por ejemplo: ruido de lugares bulliciosos/con mucha gente, exceso de ruido en el aula, etc.).]	0.83
48. I feel annoyed or uncomfortable when I hear certain sudden, unexpected, and intense noises (for example: a vacuum cleaner, thunder, etc.). [Siento molestia o incomodidad cuando escucho ciertos ruidos bruscos, inesperados e intensos (por ejemplo: una aspiradora, truenos, etc.).]	0.85
49. I feel annoyed or uncomfortable when I listen to loud noises (e.g., concerts, drills, alarm sounds, street works, etc.). [Siento molestia o incomodidad cuando escucho ruidos de volumen alto (p.ej.: conciertos, taladros, sonidos de alarmas, obras en la calle, etc.).]	0.85
50. I feel annoyed or uncomfortable when I listen to music that is not what I usually listen to. [Siento molestia o incomodidad cuando escucho una música que no es la que suele escuchar.]	0.50

**Table 5 ejihpe-15-00080-t005:** Correlations between PSRS scales.

		(1)	(2)	(3)	(4)	(5)	(6)	(7)	(8)	(9)	(10)	(11)	(12)	(13)
(1)	Total PSRS													
(2)	Pain	0.52												
(3)	Total Hypo	0.89	0.32											
(4)	Total Hyper	0.89	0.30	0.64										
(5)	Hypo-Tactile	0.73	0.29	0.84	0.50									
(6)	Hypo-Olfactory	0.71	0.25	0.80	0.51	0.60								
(7)	Hypo-Visual	0.69	0.25	0.76	0.51	0.47	0.53							
(8)	Hypo-Taste	0.64	0.18	0.73	0.47	0.47	0.48	0.51						
(9)	Hypo-Auditory	0.71	0.24	0.79	0.53	0.56	0.52	0.54	0.53					
(10)	Hyper-Tactile	0.75	0.26	0.56	0.82	0.47	0.45	0.41	0.39	0.45				
(11)	Hyper-Olfactory	0.71	0.26	0.52	0.79	0.42	0.44	0.39	0.36	0.43	0.57			
(12)	Hyper-Visual	0.65	0.18	0.48	0.73	0.33	0.35	0.41	0.42	0.40	0.51	0.50		
(13)	Hyper-Taste	0.66	0.22	0.46	0.76	0.38	0.34	0.37	0.30	0.42	0.50	0.48	0.48	
(14)	Hyper-Auditory	0.68	0.23	0.46	0.79	0.32	0.39	0.39	0.35	0.36	0.53	0.49	0.50	0.49

Note. PSRS = Pain and Sensitivity Reactivity Scale; Total Hypo = total sensory hyporeactivity; Total Hyper = total sensory hyperreactivity; All correlations are statically significant with *p* = 0.01; 1 = Total PSRS; 2 = Pain; 3 = Total Hypo; 4 = Total Hyper; 5 = Hypo-Tactile; 6 = Hypo-Olfactory; 7 = Hypo-Visual; 8 = Hypo-Taste; 9 = Hypo-Auditory; 10 = Hyper-Tactile; 11 = Hyper-Olfactory; 12 = Hyper-Visual; 13 = Hyper-Taste; 14 = Hyper-Auditory.

**Table 6 ejihpe-15-00080-t006:** Ordinal Cronbach’s Alpha, Ordinal MacDonald’s Omega, and test–retest reliability.

	Alpha	Omega	T-R Reliability
Pain	0.83	0.79	0.69
Total Sensory hyporeactivity	0.93	0.91	0.87
Hypo-Tactile	0.74	0.68	0.86
Hypo-Olfactory	0.65	0.73	0.79
Hypo-Visual	0.75	0.68	0.81
Hypo-Taste	0.71	0.62	0.74
Hypo-Auditory	0.70	0.59	0.78
Total Sensory hyperreactivity	0.92	0.90	0.81
Hyper-Tactile	0.76	0.68	0.77
Hyper-Olfactory	0.85	0.79	0.81
Hyper-Visual	0.60	0.53	0.52
Hyper-Taste	0.83	0.75	0.73
Hyper- Auditory	0.82	0.82	0.82
Total PSRS	0.95	0.94	0.86

Note. PSRS = Pain and Sensitivity Reactivity Scale; M = mean; SD = standard deviation; Hypo = sensory hyporeactivity; Hyper = sensory hyperreactivity.

**Table 7 ejihpe-15-00080-t007:** Convergent and discriminant validity scores.

		SOR					OCI-R					
		Touch	Smell	Sight	Taste	Hearing	Hoarding	Checking	Ordering	Neutralizing	Washing	Obsessing
PSRS	Pain	0.21	0.21	0.21	0.23	0.24	0.18	0.17	0.14	0.15	0.19	0.19
	Total Sensory hyporeactivity	0.42	0.36	0.36	0.37	0.40	0.43	0.39	0.38	0.44	0.37	0.43
	Hypo-Tactile	0.32	0.28	0.27	0.31	0.30	0.35	0.32	0.31	0.35	0.29	0.35
	Hypo-Olfactory	0.32	0.28	0.27	0.28	0.33	0.34	0.30	0.34	0.40	0.33	0.34
	Hypo-Visual	0.32	0.28	0.32	0.29	0.33	0.31	0.29	0.30	0.35	0.33	0.33
	Hypo-Taste	0.32	0.30	0.27	0.20	0.29	0.30	0.29	0.27	0.31	0.22	0.31
	Hypo-Auditory	0.38	0.27	0.31	0.36	0.33	0.38	0.31	0.29	0.32	0.28	0.37
	Total Sensory hyperreactivity	0.60	0.55	0.51	0.54	0.59	0.46	0.43	0.50	0.47	0.48	0.50
	Hyper-Tactile	0.54	0.42	0.43	0.40	0.44	0.42	0.42	0.48	0.44	0.44	0.48
	Hyper-Olfactory	0.44	0.58	0.36	0.41	0.39	0.36	0.33	0.36	0.36	0.38	0.35
	Hyper-Visual	0.45	0.41	0.48	0.37	0.42	0.31	0.29	0.38	0.37	0.33	0.36
	Hyper-Taste	0.45	0.36	0.31	0.59	0.38	0.32	0.28	0.33	0.34	0.32	0.33
	Hyper-Auditory	0.45	0.38	0.42	0.35	0.64	0.37	0.35	0.38	0.32	0.36	0.42

Note. PSRS = Pain and Sensitivity Reactivity Scale; OCI-R = obsessive-compulsive inventory—revised; SOR = sensory over-responsivity scale; Total Hypo = total sensory hyporeactivity; Total Hyper = total sensory hyperreactivity. All correlations are statically significant with *p* = 0.01.

## Data Availability

The raw data supporting the conclusions of this article will be made available by the authors on request. The data are not publicly available due to privacy concerns.

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
