# Peer review of "Validation of the Pain and Sensitivity Reactivity Scale in Neurotypical Late Adolescents and Adults"

_ejihpe, 2025, doi:10.3390/ejihpe15050080_

Round 1
Reviewer 1 Report
Comments and Suggestions for Authors
The present study aimed to examine the psychometric properties of a novel instrument: the Pain and Sensitivity Reactivity Scale (PSRS), in a community-based sample of Spanish adults. This study is of relevance, as a majority of instruments assessing sensory symptoms are designed to be completed by a primary caregiver. Moreover, most of these instruments do not include pain, despite its relevance and association with sensory over-responsivity and anxiety. Participants were recruited from various universities across Spain. The authors had a very large sample size to answer their aims. They observed that the PSRS had excellent psychometric properties and three factor models (Pain, Total sensory hyporeactivity, Total sensory hyperreactivity). The authors conclude that PSRS is a reliable measure for analyzing pain and sensory reactivity in neurotypical populations.
The authors wrote a very comprehensive manuscript with immense detail. However, upon reviewing the manuscript, major concerns need to be addressed.
Major concerns:
Generic
- It is unclear whether the authors are comparing outcomes as a function of sex or gender. Sex is a biological variable, while gender is a social and cultural construct that is self-reported. Please clarify and correct in the manuscript for coherence.
Title
- The title of the manuscript is “Validation of the Pain and Sensitivity Reactivity Scale in neurotypical late adolescents”. However, the sample includes late adolescents AND adults. Moreover, it is unclear how the authors determined that their sample was “neurotypical”. The authors clearly state that this is a community-based sample. Therefore, the title of the manuscript needs to be corrected.
Introduction
- The introduction is very lengthy with an unclear structure that is hard to follow as a reader. The authors discuss anxiety, OCD, and abdominal pain, which are not the focus of the manuscripts results. I suggest the authors to cut down on the introduction to make it more coherent with the methods and results.
- Redundant and repetitive sentences are found throughout the introduction. Lines 125-131 are sentences that were written in the same way in Lines 82-88. Moreover, Lines 144-151 are direct copy-pasted sentences from 134-142. I highly suggest the authors review their introduction in its entirety for coherence and relevance to their study.
Methods
- Section 2.1 Data and Sample, is not considered “methods”, but “results” and should be moved accordingly.
- Section 2.3 “Development and Content Validity of the New Instrument” is almost an exact replication from the authors’ previous work (Martínez-González et al., 2024a). I recommend that the authors reduce this section to make it more coherent, and merge it with the previous subsection “Pain and sensitivity reactivity scale (PSRS)”.
- Line 232. The self-report version was used in the present study. However, it is unclear what adjustments were made between the presented version and the caregiver version from the authors’ previous publication (Martínez-González et al., 2024a).
- Section 2.4 Procedure. There are ethical considerations that the authors seem to have missed to report. What was the age limits from which the authors recruited their community-based sample? Where did you recruit late-adolescents? Why were young adolescents not included? What is the rationale to make the cut-off of late adolescents 19 (according to the results)? How were participants recruited? Was there an IRB approval for this study?
- Figure 1 is an exact replication from the authors’ publication: Martínez-González et al., 2024a. I highly recommend removing the figure, or properly referencing it, if possible with the journal, and if needed for the manuscript.
Results
- Line 387-389. It is unclear why the results of factorial invariance as a function of sex is mentioned in this section.
- Table 4. It is unclear what is presented in this table. Are the number presented the correlation coefficients? Please clarify in the title or legend of the table.
- Line 419-435. This should be a separate section, as it pertains to the Aim 4. Moreover, the numbers do not seem to match the Table 6, and Table 6 seems to be missing columns for the subscales of the OCI-R. The authors should review this section.
- Section 3.7. Was the invariance as a function of sex and gender conducted with item 17 removed (mentioned in Line 387-389)? Please clarify.
Discussion
- The authors should expand on the limitations of the study.
- Line 607-616. This is a conclusion section that should be to the last section of the manuscript. However, the authors mention in line 611 “selective diet, and gastrointestinal symptoms in autism and other disorders”. It is unclear the relevance of this in the current manuscript. How does this instrument help in measuring the relationship between pain and reactivity with GI-symptoms, if GI symptoms were not a variable measured in the sample. I acknowledge that this instrument can be used in multiple populations, but the authors’ focus on GI-related disorders is not necessary.
Minor concerns:
Abstract
- In the conclusion of the abstract, the authors state (Line 29-30): “PSRS is a reliable measure for analyzing pain and sensory reactivity in neurotypical population.” However, this is very strong claim to present, as this instrument has only been tested in a sample of autistic people (Martinez-Gonzalez, 2024a) and the community presented in the current manuscript. I suggest switching the word “is” for “may be”.
Methods
- Line 292: Can the authors clarify what were the “study procedures”. To these allude to the “Measurements”? If so, I suggest changing “study procedures” to “study questionnaires”.
- Line 293. Were adolescents also recruited from the universities?
- Line 293 and 302. The authors first mention recruitment from universities (Line 293), but then refer to college students (line 302). Please clarify.
Results
- It is unclear why the authors separated Table 2a and Table 2b as “sub-tables”. Numbering of tables should be consecutive (1, 2, 3, etc.)
- Supplementary Tables 1 & 2. Please clarify in table legends what t, p and d represent.
Discussion
- Line 574-576. The sentence is redundant, as it is similar to Lines 554-557.
- Line 590. The sentence is redundant, as it is similar to Lines 585-587.
- Line 625. The first sentence only mentions late adolescents, but the study also included adults. Please be cohesive throughout the manuscript.
Please see comments above for major revisions on structure.
The authors should also review their manuscript in its entirety for any minor orthographic mistakes, especially on verb tenses (using the present versus the past tense).
Author Response
Generic
- It is unclear whether the authors are comparing outcomes as a function of sex or gender. Sex is a biological variable, while gender is a social and cultural construct that is self-reported. Please clarify and correct in the manuscript for coherence.
Response: Data are analysed on the variable function of gender. The term sex has been removed.
Title
- The title of the manuscript is “Validation of the Pain and Sensitivity Reactivity Scale in neurotypical late adolescents”. However, the sample includes late adolescents AND adults. Moreover, it is unclear how the authors determined that their sample was “neurotypical”. The authors clearly state that this is a community-based sample. Therefore, the title of the manuscript needs to be corrected.
Response: The word adults has been included in the title.
Introduction
- The introduction is very lengthy with an unclear structure that is hard to follow as a reader. The authors discuss anxiety, OCD, and abdominal pain, which are not the focus of the manuscripts results. I suggest the authors to cut down on the introduction to make it more coherent with the methods and results.
Response: Thank you for your comment. The content that talks about the results in clinical population has been removed.
- Redundant and repetitive sentences are found throughout the introduction. Lines 125-131 are sentences that were written in the same way in Lines 82-88. Moreover, Lines 144-151 are direct copy-pasted sentences from 134-142. I highly suggest the authors review their introduction in its entirety for coherence and relevance to their study.
Response: Duplicate texts that were incorrectly copied in the edition of the manuscript according to the journal version have been removed.
Methods
- Section 2.1 Data and Sample, is not considered “methods”, but “results” and should be moved accordingly.
Response: The change has been made
- Section 2.3 “Development and Content Validity of the New Instrument” is almost an exact replication from the authors’ previous work (Martínez-González et al., 2024a). I recommend that the authors reduce this section to make it more coherent, and merge it with the previous subsection “Pain and sensitivity reactivity scale (PSRS)”.
Response: Thanks for the suggestion. A reduction of the text has been made.
- Line 232. The self-report version was used in the present study. However, it is unclear what adjustments were made between the presented version and the caregiver version from the authors’ previous publication (Martínez-González et al., 2024a).
Response: More information about the adaptation process is added to the manuscript.
There were no changes in the items between the caregiver version of the PSRS and the self-report scale. The grammar was adapted to the first person singular for each of the items in the self-report version of the PSRS.
- Section 2.4 Procedure. There are ethical considerations that the authors seem to have missed to report. What was the age limits from which the authors recruited their community-based sample?
It has been decided to analyze the young adult population and late adolescence, which according to the World Health Organization is between 15 and 19 years old.
- Where did you recruit late-adolescents?
They were recruited in secondary education centers
Late adolescents who were not yet in university were selected from two secondary education centers in Alicante and Murcia.
- Why were young adolescents not included?
This is an excellent idea that we plan to explore in the future
- What is the rationale to make the cut-off of late adolescents 19 (according to the results)?
From the age of 15 according to the World Health Organization
- How were participants recruited?
Contacting the educational centers
- Was there an IRB approval for this study?
Information about the ethics committee is provided in the Ethical Considerations section.The present study was approved by the Ethics Committee (reference number: UA-2019-10-04, approval date: March 27, 2020).
- Figure 1 is an exact replication from the authors’ publication: Martínez-González et al., 2024a. I highly recommend removing the figure, or properly referencing it, if possible with the journal, and if needed for the manuscript.
Figure 1 has been removed.
Results
- Line 387-389. It is unclear why the results of factorial invariance as a function of sex is mentioned in this section.
Response: The term has been determined as gender
- Table 4. It is unclear what is presented in this table. Are the number presented the correlation coefficients? Please clarify in the title or legend of the table.
Response: A legend has been included for each subscale of the PSRS
- Line 419-435. This should be a separate section, as it pertains to the Aim 4. Moreover, the numbers do not seem to match the Table 6, and Table 6 seems to be missing columns for the subscales of the OCI-R. The authors should review this section.
Response: It has been included as a validity section. The figures in the text have been read. Table 6 contains all the variables of the OCIR
- Section 3.7. Was the invariance as a function of sex and gender conducted with item 17 removed (mentioned in Line 387-389)? Please clarify.
Response: Effectively, factorial invariance as a function of gender and age was examined and item 17 was removed. This is indicated in the text
Discussion
- The authors should expand on the limitations of the study.
Response: Thanks for the comment. One more limitation has been added
Finally, the fourth limitation is that there is not a large sample of participants for the evolutionary periods of late adolescence and early adulthood.
- Line 607-616. This is a conclusion section that should be to the last section of the manuscript. However, the authors mention in line 611 “selective diet, and gastrointestinal symptoms in autism and other disorders”. It is unclear the relevance of this in the current manuscript. How does this instrument help in measuring the relationship between pain and reactivity with GI-symptoms, if GI symptoms were not a variable measured in the sample. I acknowledge that this instrument can be used in multiple populations, but the authors’ focus on GI-related disorders is not necessary.
Response: This information has been deleted
Minor concerns:
Abstract
- In the conclusion of the abstract, the authors state (Line 29-30): “PSRS is a reliable measure for analyzing pain and sensory reactivity in neurotypical population.” However, this is very strong claim to present, as this instrument has only been tested in a sample of autistic people (Martinez-Gonzalez, 2024a) and the community presented in the current manuscript. I suggest switching the word “is” for “may be”.
Response: It has changed.
Methods
- Line 292: Can the authors clarify what were the “study procedures”. To these allude to the “Measurements”? If so, I suggest changing “study procedures” to “study questionnaires”.
- Line 293. Were adolescents also recruited from the universities?
- Line 293 and 302. The authors first mention recruitment from universities (Line 293), but then refer to college students (line 302). Please clarify.
Response: In the procedure section it has been reported that the participants were university students (over 17 years old) and high school teenagers who were 16 years old)
Results
- It is unclear why the authors separated Table 2a and Table 2b as “sub-tables”. Numbering of tables should be consecutive (1, 2, 3, etc.)
The suggested change has been made
- Supplementary Tables 1 & 2. Please clarify in table legends what t, p and d represent.
These variables have been clarified

Reviewer 2 Report
Comments and Suggestions for Authors
Thank you for the opportunity to review this manuscript. I believe that this study validating the Pain and Sensitivity Reactivity Scale in neurotypical adolescents makes a significant contribution to the literature, especially given that the scale has been previously validated in neurodivergent individuals. This manuscript has merit, and would likely benefit from some suggested improvements which I've listed by section below.
Introduction:
- Page 2, starting at line 58, there is a lengthy section discussing the links between ASD, OCD, RRBs, and reactivity. While this content provides some important context to the relation between multiple conditions and reactivity, I believe that this paragraph could be condensed given that the current study investigates a non-clinical population.
- Page 2, line 97 refers to "high-functioning" autism. There has been a push to move away from this terminology and instead refer to the level of support needed (e.g., "level 1 autism or requiring minimal support". For more information, see Autism Awareness here Why we should stop using the term “high functioning autism” | Autism Awareness Australia.
- I appreciated the point on page 3 about estimating variability in sensory reactivity in the general population and believe this is a strong case for this article!
- Page 3, line 146-147 repeats what is said in line 137-138 making it redundant. I'm not sure that the paragraph starting on line 144 is entirely necessary either given that the description in the paragraph prior is sufficient for the purpose of this study.
Methods and Materials
- In the introduction, it is mentioned that there are several existing measures for reactivity but only the SORS was included in the current study, why is that? Are the others not self-report? This is hinted at in the introduction but not stated explicitly.
- This is small, but the measures state that the PSRS is a Likert scale but technically it would be Likert-type given that it is 4-point.
- I was unclear throughout the methods which language that the study was conducted in - English, Spanish, or both?
Results
- Related to demographics, if it was collected it would be beneficial to report sex at birth and gender of patients.
- Related to the PSRS questionnaire itself, I was wondering if it was professionally translated by a medical interpreter from one language to the other? A few of the items in English were worded a bit oddly including number 6 (a clearer reference may be "blood test" or "vaccine" as opposed to analysis in English), numbers 30, 32, 36, 37, 38, 39, 43, 45 (they say "I feel discomfort or discomfort" which is redundant).
- Relative to convergent and discriminant validity, it appeared that all correlations were significant, though their levels varied. For discriminant validity, would it not be more beneficial if correlations were not significant for the less-related scales? This appears to support convergent, but possibly not discriminant validity.
Discussion
- The first two paragraphs of the discussion felt more like they belonged in the introduction (providing rationale for conducting the study). It felt like the discussion could have started with the third paragraph.
- Page 18, line 560-561, authors state that results indicate that hyporeactivity is independent of compulsive behaviors - but do they? The correlation was still significant, so I'm not sure I agree with that interpretation.
- On page 19, the authors note prior research suggesting that females are more sensitive to pain than males, but the nuance in this issue around pain reporting could be worth discussing as well (i.e., men are more likely to under-report pain as they are expected to be more stoic).
- See Templeton, K. J. (2020). Sex and gender issues in pain management. JBJS, 102(Suppl 1), 32-35. for an example
Conclusions
- In the conclusions, authors state that understanding sensitivity could improve individualized treatment and I would have liked to learn more about how that could be done in the discussion.
Author Response
Introduction:
- Page 2, starting at line 58, there is a lengthy section discussing the links between ASD, OCD, RRBs, and reactivity. While this content provides some important context to the relation between multiple conditions and reactivity, I believe that this paragraph could be condensed given that the current study investigates a non-clinical population.
Response: Thank you for your comment. The content that talks about the results in clinical population has been removed.
- Page 2, line 97 refers to "high-functioning" autism. There has been a push to move away from this terminology and instead refer to the level of support needed (e.g., "level 1 autism or requiring minimal support". For more information, see Autism Awareness here Why we should stop using the term “high functioning autism” | Autism Awareness Australia.
Response: This information has been rectified
- Page 3, line 146-147 repeats what is said in line 137-138 making it redundant. I'm not sure that the paragraph starting on line 144 is entirely necessary either given that the description in the paragraph prior is sufficient for the purpose of this study.
Response: Duplicate texts that were incorrectly copied in the edition of the manuscript according to the journal version have been removed.
Methods and Materials
- In the introduction, it is mentioned that there are several existing measures for reactivity but only the SORS was included in the current study, why is that? Are the others not self-report? This is hinted at in the introduction but not stated explicitly.
Response: Scales that do not have a self-reported version are indicated. The SOR is the only self-reported scale that has been able to analyze its psychometric properties in Spain. For that reason, only this scale is included in the study.
- This is small, but the measures state that the PSRS is a Likert scale but technically it would be Likert-type given that it is 4-point.
Response: The PSRS is a 4-point Likert scale.
- I was unclear throughout the methods which language that the study was conducted in - English, Spanish, or both?
Response: This information has been added to the procedure section.
The assessment protocol was only written in Spanish.
Results
- Related to demographics, if it was collected it would be beneficial to report sex at birth and gender of patients.
Response: This information is added to the results section.
- Related to the PSRS questionnaire itself, I was wondering if it was professionally translated by a medical interpreter from one language to the other? A few of the items in English were worded a bit oddly including number 6 (a clearer reference may be "blood test" or "vaccine" as opposed to analysis in English), numbers 30, 32, 36, 37, 38, 39, 43, 45 (they say "I feel discomfort or discomfort" which is redundant).
Response: The scale is of Spanish origin. It has been reported in more detail in the procedure section. In the Spanish language at the medical level, feeling discomfort does not have to be the same as discomfort.
- Relative to convergent and discriminant validity, it appeared that all correlations were significant, though their levels varied. For discriminant validity, would it not be more beneficial if correlations were not significant for the less-related scales? This appears to support convergent, but possibly not discriminant validity.
Response: Thanks for the contribution. In this study, two main measures have been used. To analyze the convergent validity of the SOR and the discriminant validity with the OCI-R. Although we already argued in the introduction that moderate correlations can appear between repetitive behavior, analyzed in this study with the OCI-R and sensory reactivity. Actually, the OCIR is a measure that discriminates on the pain variable of the PSRS, because its correlations are very low.
Discussion
- The first two paragraphs of the discussion felt more like they belonged in the introduction (providing rationale for conducting the study). It felt like the discussion could have started with the third paragraph.
Response: The two opening paragraphs have been deleted.
- Page 18, line 560-561, authors state that results indicate that hyporeactivity is independent of compulsive behaviors - but do they? The correlation was still significant, so I'm not sure I agree with that interpretation.
Response: That statement is rectified and this text is added.
These results indicate that sensory hyporeactivity and repetitive behavior have a smaller association.
- On page 19, the authors note prior research suggesting that females are more sensitive to pain than males, but the nuance in this issue around pain reporting could be worth discussing as well (i.e., men are more likely to under-report pain as they are expected to be more stoic).
See Templeton, K. J. (2020). Sex and gender issues in pain management. JBJS, 102(Suppl 1), 32-35. for an example
Response: We appreciate your contribution and it is included in the discussion section.
In the same way, it must be considered that there may be a bias of the male informant. Due to social desirability, men may not report pain tolerance (Templeton, 2020).
Conclusions
- In the conclusions, authors state that understanding sensitivity could improve individualized treatment and I would have liked to learn more about how that could be done in the discussion.
Response: We appreciate your interest, but this article focuses on the psychometric properties of the PSRS. In the future, it would be interesting to analyze the usefulness of the instrument for the curricular adaptations of students with and without special educational needs, to analyze the effectiveness of the interventions, etc.

Round 2
Reviewer 1 Report
Comments and Suggestions for Authors
The authors addressed all of my comments. However, a minor comment remains.
- The authors removed the word “sex” across the manuscript, and replaced it with "gender" according to their analyses. Sex is still mentioned in line 382. Please correct.
Author Response
The word change has been made
Reviewer 2 Report
Comments and Suggestions for Authors
Authors have addressed all concerns.
Author Response
Thank you for your support